# Transgenerational Perpetuation of CHS Gene Expression and DNA Methylation Status Induced by Short Oligodeoxynucleotides in Flax (*Linum usitatissimum*)

**DOI:** 10.3390/ijms20163983

**Published:** 2019-08-16

**Authors:** Magdalena Dzialo, Jan Szopa, Agata Hnitecka, Magdalena Zuk

**Affiliations:** 1Department of Genetic Biochemistry, Faculty of Biotechnology, Wroclaw University, Przybyszewskiego 63, 51-148 Wroclaw, Poland; 2Linum Foundation, pl. Grunwaldzki 24a, 50-363 Wroclaw, Poland; 3Department of Genetics, Plant Breeding and Seed Production, Wroclaw University of Environmental and Life Sciences, pl. Grunwaldzki 24a, 50-363 Wroclaw, Poland

**Keywords:** chalcone synthase, chromatin accessibility, epigenetic inheritance, genetic engineering, methylation, ODNs

## Abstract

Over two decades ago, short oligodeoxynucleotides (ODNs) were proven to be an effective and rapid technique for analysis of gene function without interference in the plant genome. Our previous research has shown the successful regulation of chalcone synthase (CHS) gene expression in flax by ODN technology. The CHS gene encodes a pivotal enzyme in flavonoid biosynthesis. The manipulation of its transcript level was the result of the specific methylation status developed after treatment with ODNs. In further analysis of the application of oligodeoxynucleotides in plants, we will focus on maintaining the methylation status induced originally by ODNs homologous to the regulatory regions of the CHS gene in flax. This article reports the latest investigation applied to stabilization and inheritance of the epigenetic marks induced by plants’ treatment with ODNs. The methylation status was analyzed in the particular CCGG motifs located in the CHS gene sequence. Individual plants were able to maintain alterations induced by ODNs. In order to confirm the impact of methylation marks on the nucleosome rearrangement, chromatin accessibility assay was performed. The perpetuation of targeted plant modulation induced by ODNs exhibits strong potential for improving crops and intensified application for medicine, nutrition and industry.

## 1. Introduction

A considerable number of studies have confirmed that epigenetic mechanisms play a crucial role in the ability of plants to adapt to the environment during stress conditions [1,2,3]. Heritable epigenetic variability can lead to physiological, morphological and ecological changes. Alterations of the epigenome can improve the plant features valuable for agronomical reasons [4]. Precise manipulation of the DNA methylation status enables one to control chromatin condensation, and thus gene expression. The assessment of detailed mechanisms that regulate the epigenetic changes under particular stress factors could provide information about the characteristic epigenetic marks that lead to adaptation in encountered stress conditions [5].

Epigenetic modifications induced under stress factors can be reversed and epigenomic status can return to the initial state, when the stress conditions are no longer present. However, the latest findings have shown that some of these changes can be preserved as stress memory leading to inheritance of the epigenetic status and thus enhancement of the adaptable ability of the progeny to the environment [5]. In comparison to mammalian cells, in plant cells, massive erosion of the epigenetic marks during gametogenesis occurs [6]. Particular short RNA molecules are assumed to crucially contribute to the inheritance of the epigenetic changes. Due to the RNA-dependent DNA methylation, the methylation implemented to the genome of parental cells can be maintained after cell division. The maintenance of the DNA methylation after replication is the effect of the feed-forward maintenance of de novo methylation in plants. The essence of this process is the establishment of two processes. The first mechanism involves methylation in the CpG and CpHpG contexts (H indicates cytosine, adenine or thymine) through methyltransferase 1 (MET1) and chromomethylase 3 (CMT3) in the newly synthesized DNA strand. However, cytosine methylation is also maintained through RNA-dependent DNA methylation by short RNA molecules stimulated by DNA methylation. The occurrence of these two processes leads to independence of the DNA methylation mechanism in the plant cell [7].

The latest research has shown that apart from environmental stress, epigenetic changes may occur as an effect of short oligodeoxynucleotides’ (ODNs/OLIGOs) activity. The mechanism of ODNs is based on the development of particular short non-coding RNAs, which may lead to the elevated accumulation (RNA activation—RNAa) or degradation (RNA interference—RNAi) of the gene transcript and the RNA-directed DNA methylation (RdDM) [8,9].

In our previous work [9], we demonstrated the effectiveness of the short oligodeoxynucleotides in the induction of epigenetic changes in flax. The plant infiltration with ODNs complementary to the regulatory regions (ODN1 to 5’UTR, ODN6 to intron, ODN11 to 3’UTR) of the chalcone synthase (CHS) genes *LuCHS6* and *LuCHS7* led to changes in the CHS genes’ expression. The alteration of the *CHS* transcript levels was assumed to be triggered by the changes in the DNA methylation. The identification of the 5-methylcytosine (5-mC) in the CCGG motifs located in the CHS genes made it possible to distinguish the stably unmethylated motifs, independently of the analyzed plant (*LuCHS6*/*LuCHS7*: −232/−, +217/−, +1606/+1156) and variable methylated motifs (*LuCHS6*/*LuCHS7*: +996/+552, +1219/+775, +1273/+829). The in silico analysis of nucleosome formation energy also indicates the alteration of nucleosome position upon the changes in the methylation status of *LuCHS6*. The preliminary examination of this theoretical presumption suggests that the increased level of methylation in the +1273 CCGG motif of the transgenic plants led to the visible changes in the accessibility of DNA for the restriction enzymes. Thus, the obtained results suggested the variable susceptibility of DNA upon methylation to the interaction with the particular effector proteins (e.g., transcription machinery) [9].

The subject of our research was the linseed variety of common flax *Linum usitatissimum* L.–Linola. Linseed is widely known for its biomedical applications due to an appropriate ratio of unsaturated fatty acids in the linseed-derived oil (the ω-6:ω-3 ratio is 1:4) [10], as well as high content of phenylpropanoid compounds (flavonoids, lignans and phenolic acids) [11,12] and fiber [13]. Flavonoids present in the flaxseed are popular in the context of health-promoting properties, including antioxidative [14] and antimicrobial activity [15]. Recent findings indicate a preventive or therapeutic role of flavonoids from flax in metabolic disorders [16] and carcinogenesis [17,18].

Chalcone synthase (CHS) catalyzes the first step of flavonoid biosynthesis [19]. In many plant species, there occur more than one gene encoding CHS (multigenomic family). It was demonstrated that there are 5 CHS genes (Phytozome ID)—*LuCHS3* (Lus10041508), *LuCHS4* (Lus10023670), *LuCHS5* (Lus10011746), *LuCHS6* (Lus10033717), *LuCHS7* (Lus10031622)—and 4 CHS-like genes (Phytozome ID): *LuCHS1* (Lus10039904), *LuCHS2* (Lus10042388), *LuCHS8* (Lus10026286), *LuCHS9* (Lus10002187) in the genome of *Linum usitatissimum* L. [20]. The transcripts derived from the 5 mentioned CHS genes were also confirmed by our group [9,21]. Moreover, our findings indicated the presence of 2 additional CHS genes, named in order: *LuCHS10* and *LuCHS11* (recognized fragment of *LuCHS11* sequence is presented in Appendix A). Initially, they were only suspected to be transcriptional variants of *LuCHS3* and *LuCHS4,* respectively. However, the in silico analysis showed the diverse localization of these genes on the chromosomes. The systematized names, numbers in databases and genomic localization are presented in Table 1. Apart from the fact that CHS genes encode the key enzyme in flavonoid formation, they display a susceptibility to the response to stress factors and can be specifically regulated under particular environmental conditions. The expression of the chalcone synthase genes may also depend on the plant developmental stage and tissue specificity. In our research, we focused on two CHS genes: *LuCHS6* and *LuCHS7*. The genomic sequences of these genes are similar to the CHS-A encoding gene from *Petunia* x *hybrida*, used for the generation of genetically modified (GM-CHS) flax, which served as the model for changes in CHS gene expression in our initial research.

The main purpose of our research was the monitoring of changes induced by the treatment with ODNs after a time longer than 48 h since the beginning of incubation. The plants from subsequent generations derived from ODN-treated flax were analyzed in the context of the *LuCHS6* and *LuCHS7* gene expression and methylation of CCGG motifs, in order to verify the perpetuation of the induced changes. With a view to the relative susceptibility to ODNs’ degradation by nucleases, modification of the oligodeoxynucleotide molecules was applied—by methylation of cytosines present in the ODN sequence (met) and by the replacement of oxygen by a sulfur at the phosphodiester bond (thiophosphate–pto). Stability of the epigenetic changes induced by the modified ODNs was also analyzed in the next generation of plants. Moreover, the chromatin accessibility assay has shown the differences between the control (Linola) and plants with stable epigenetic modification (W92.40). Performed experiments contributed to the understanding of the processes involved in the introduction of exogenous oligonucleotides into plant cells.

## 2. Results

### 2.1. Heredity of CHS Gene Expression in the F1, F2 and F3 Generations of ODN-Treated Flax.

The nomenclature of the CHS gene isoforms *CHS1* (Phytozome database, Lus10031622) and *CHS2* (Phytozome database, Lus10033717) published in our previous work [9] will be followed in the present paper by the numbers used by [20], sc. *LuCHS6* (Phytozome database, Lus10033717) and *LuCHS7* (Phytozome database, Lus10031622).

The level of the mRNA derived from both studied isoforms, *LuCHS6* and *LuCHS7*, was analyzed in the F1 and F2 generation of plants treated with ODNs. In our previous study, the series of ODN sequences homologous to *CHS* was investigated. The oligodeoxynucleotides directed to the regulatory regions, 5’UTR-ODN1, intron-ODN6 and 3’UTR-ODN11 have shown significant modulation in the CHS gene expression. For ODN1 and ODN11, overexpression of the endogenous CHS gene was observed. Alternatively, for ODN6, the gene repression was noted for the studied gene. These three ODNs were selected for further analyses. Table 2 presents the maintenance of the modulation of CHS gene expression induced by ODNs in the next generations (F1–F3).

A part of seeds obtained from the F0 plants treated with ODNs were introduced into the in vitro culture in order to select the lines that maintained the initially induced changes in the *CHS* expression. The percentage proportion of the plants that maintained the tendency of the initial modulation to all analyzed plants was as follows: ODN1-71%, ODN6-5%, ODN11-86%, where the studied groups varied between 7 to 20 individuals in the F1 plant generation. The most plants with the stable modification in the *CHS* gene were noted for the ODN11, whereas the least for ODN6. Seeds of selected plants that maintained the generated characteristics served to obtain the next generation (F1) under field conditions.

Seeds collected from the F1 generation of plants were also partially introduced to the in vitro conditions. The total gene expression of *LuCHS6* and *LuCHS7* (the sum) was determined in the F2 flax. Among the series of plants, individuals that presented a tendency of *CHS* expression similar to the parental generation were selected. The most plants that perpetuated the alteration in the CHS gene expression was observed according to the ODN11 treatment (75%—6 of 8 ODN11 treated plants). A minor percentage (33%) of the ODN1 treated plants maintained in the F2 generation induced primarily changes. The repression noted after ODN6 in the F0 and F1 plants was not observed in the F2 generation and the massive overexpression in those plants was investigated.

The F3 field generation of plants treated with ODNs was also determined in the context of the maintenance of the induced changes in the CHS transcript level. A decreased percentage of modulation maintenance was observed for ODN1 (38%) and ODN11 (6%) in relation to the previous generations. In contrast, the number of plants that perpetuated repression of the CHS gene was the highest among the F3 generation of ODN-treated plants and the maintenance of the induced changes reached 61%.

### 2.2. Heredity of Epigenetic Changes in the F1 Generation of ODN-Treated Flax Cultivated in the Field

It has been repeatedly demonstrated that CHS gene expression may be induced in plants under stress conditions such as UV light, and bacterial or fungal infection; therefore it was decided to check during plant growth, as the environmental stress present in the field will affect the presence of induced epigenetic changes. The seeds from F0 generation plants treated with ODNs were cultivated in the experimental field. During growth of F1 plants in natural conditions, individual young plants (6-week-old plants in the middle of the vegetative growth period) were selected (and harvested) for testing from a pool of plants in order to determine the maintenance of the induced features.

#### 2.2.1. CHS Gene Expression in the Selected Plants

Among plants cultivated in the field, the highest percentage of the flax treated ODN1 presented the primarily induced modulation of *CHS* expression (75%) Only 33% of the analyzed F1 generation of plants incubated with ODN11 possessed overexpression of the CHS gene. For ODN6, none of the analyzed plants presented expected repression of the studied gene. In Figure 1, the CHS gene expression for selected individuals is presented.

#### 2.2.2. Pattern of DNA Methylation in Variable CCGG Motifs of the CHS Gene

For the plants presented in Figure 1, the methylation status of the variable CCGG sites was evaluated (Figure 2A–D). In particular F1 individuals, methylation of CCGG motifs was mostly similar to the profiles observed in our previous study [9]. The methylation status of F1 control plants resembled the results for F0 control plants. However, at the +1273 site, a reduced percentage of demethylation (22.5%) for the F1 generation was noted (Figure 2A), while for the F0 generation, the relevant status measured 33.9%. For ODN1, the percentage of CCmGG methylation at the +996 site was similar to the methylation level observed in the plants after treatment. The methylation profile at the +1273 site for plants modified by ODN1 was corresponding between generations; hence, in the F1 generation an increased methylation status (by 15.8%) was observed (Figure 2B). According to ODN6, despite the observed reversal of the tendency of change in CHS gene expression, the percentage of CCmGG at the +996 motif was stable, as observed after 48 h after incubation. Also at the +1273 site, the tendency of change was maintained and a minor increase in demethylation was observed (Figure 2C). For ODN11, significant stable methylation of internal cytosine CCmGG was noted in the +996 and +1273 motifs, which correlates with primarily induced ODN modifications (Figure 2D).

### 2.3. Activity of ODNs Modified by Methylation (Met) and Thiophosphate (Pto)

Susceptibility of the oligodeoxynucleotide sequences for the nucleolytic digestion limits the lifetime of ODNs. In order to extend the time of action of short oligonucleotides, two modifications of the nucleotides included in the sequences were performed: 1. methylation of all cytosines in the particular ODN (met) 2. substitution with a sulfur atom instead of one of the nonbridging oxygen atoms in the phosphate backbone (pto).

#### 2.3.1. CHS Gene Expression Studies 10 Days after ODN Incubation

The impact of the unmodified, and modified 1. met and 2. pto ODNs on the gene expression of both studied isoforms is presented in Figure 3. In order to determine whether the effect obtained by modified ODNs lasts longer, the gene expression was investigated in the material harvested after at least 10 days after the incubation with oligodeoxynucleotides. The value of *CHS* expression after ODN1 and ODN11 treatments was at a similar level (RQ ~2), regardless of the sequence modification, in comparison to the control. However, the ODN6 modified by methylation and thiophosphate triggered repression more prominently than unmodified ODN6, respectively by 41% and 63% in comparison to the control.

#### 2.3.2. Pattern of DNA Methylation in Variable CCGG Motifs of the CHS Gene after Treatment by Unmodified and Modified Oligos

In order to determine a specific pattern of the CHS gene methylation, crucial -CCGG- motifs in the CHS gene sequence were analyzed. As presented in the previous work, among the studied sites, the stable demethylated and highly variable in methylation -CCGG- were observed. Stable lack of cytosine methylation was observed in the following regions: 5’UTR (−232), non-coding (+217) and exon 2 (+1606), which were basically unchanged in comparison to the control. Variable sites were noted only in the coding region (+996, +1219, +1273). The figures show the percentage of lack of methylation, methylation of single inner cytosine and both cytosines. Since the stable sites do not differ in the cytosine methylation profile between studied plants, only results for variable CCGG sites are presented.

The analysis of the CCGG sites for unmodified and modified sequences of ODN11 is presented in Figure 3. Similarly as in Figure 4, the experiment was performed in the material harvested 10 days after the moment of incubation with oligodeoxynucleotides. After 10 days, the original sequence ODN11 did not lead to maintaining the changes in the methylation profile. The data presented by [9], shown in Figure 4D, indicate the potent increase in CmCGG methylation after 48 h in unmodified ODN11, in comparison to the control. However, at 10 days after the flax incubation with the ODN11, the methylation profile became similar to that observed in the control. In case of ODN11 met and ODN11 pto, the perpetuation of initially induced cytosine methylation was observed (Figure 4). In ODN11 met, a potent increase of CmCGG was observed in all three variable sites +996, +1219, +1273, respectively by 19.3%, 23.0% and 13.8%, whereas for ODN11 pto in comparison to the control a higher percentage was noted in the site +1219, by 15.0% of single cytosine methylation and 2.01% of double cytosines methylation. An increase in CmCmGG methylation after ODN11 pto was also observed at the +996 site (by 8.1 %).

#### 2.3.3. Heredity of CHS Gene Expression in the F1 and F2 Generations of Modified ODN-Treated Flax Cultivated In Vitro

The flax plants treated with modified ODNs were cultivated in the experimental field. A part of the seeds obtained were sterilized and introduced to the in vitro culture. The stabilization of the modulation of CHS gene expression by modified oligodeoxynucleotides in the F1 generation of plants was investigated and the results are presented in Table 3. For ODN1 met, 4 out of 5 (80%) studied plants maintained the overexpression of *CHS* in comparison to the control (set as 1).

Regarding ODN6, the modification of oligonucleotide sequences via methylation made it possible to select one individual (out of five analyzed plants) with a repressed level of *CHS* expression (other four plants demonstrate similar too control level of *CHS* expression). For ODN11, both met and pto modifications were investigated in the next generation. Out of 7 analyzed plants, 6 presented transmission of the primarily induced modification in the chalcone synthase gene expression (data not presented).

The F2 generation plants obtained after treatment with modified ODNs were analyzed in order to assess maintenance of the induced changes in chalcone synthase gene expression. In comparison to the F1 generation, the number of analyzed plants was higher (25–30 individual plants for each ODN, except ODN6 with 6 plants). Despite the percentage of maintaining induced changes being lower, the results of F2 analysis were more significant.

### 2.4. Chromatin/DNA Accessibility Assay

For the Linola and genetically modified GM-CHS plants (W92.40), the chromatin/DNA accessibility assay was conducted. The chromatin isolated from in vitro cultured plants was digested with particular restriction enzymes: AatII and PvuI (particular sites for digestion were localized in the CHS gene, around the +1273 CCGG motif). Subsequently, DNA derived from the isolated chromatin was also incubated with restriction enzymes. The nucleic acid was purified and the effect of DNA digestion was assessed by real-time PCR at the methylation CCGG site (+1273). Figure 5 presents the relative quantity of the product obtained for the motif +1273 for undigested control, chromatin and DNA digested with restriction enzymes for Linola and GM-CHS plants. The results of this study have shown that between Linola and genetically modified plants, significant differences in the quantity of the product were obtained. For the GM-CHS flax with the stable methylation in the +1273 CCGG motif an increased level of the product was noted; thus, for transgenic plants, less accessibility of chromatin for restriction enzymes in comparison to the non-transgenic Linola was assumed. No relevant changes were observed in the +1273 motif for the DNA accessibility between the Linola and GM-CHS plants.

## 3. Discussion

Genetic engineering methods allow precise modification of the phenotype by introducing changes into the sequence of the plant genome. A well-known and commonly used method of plant genetic modification is *Agrobacterium*-mediated plant transformation. An important aspect of this method is the possibility to acquire stable transformants. Unfortunately, the practical application of transgenic plants for industrial purposes is limited in most European countries, due to restrictive legal regulations. The main reason is the lack of trust of potential GM plant consumers in relation to the harmless impact on people and animals. Another issue would be the risk of the disturbance to the natural environment by transgenic crops. The lack of public acceptance of GM crops encourages scientists to search for new methods that enable induction of organism variability without the need to change the genome [22].

Recently, a lot of attention has been devoted to methods based on the use of site-specific nucleases (SSNs), including TALENs, ZFNs and CRISPR/Cas9 [23]. Although the principle of SSN technology is based only on the editing of endogenous genes, the legal regulations regarding this method are not entirely resolved. Some genome editions can be considered as mutations and thus might be controlled by legal restrictions that apply to GMO [24]. Methods that will allow the use of the natural cell genetic repertoire are eagerly in demand. Epigenetic changes are modifications that take place in nature, mainly during the plant adjustment to constantly changing environmental conditions. In addition, the epigenetic DNA pattern can be maintained during cell division and be inherited [7,25]. Therefore, epigenetic modifications have a large potential for application in the phenotype variability of plants since they are naturally occurring changes.

Plants are considered as the best model for epigenetic regulation research. All the epigenetic mechanisms described so far occur in plants. In the plant genome, methylation in the 5th position of the cytosine in a DNA chain is a significant epigenetic modification. In plants, 5-methylcytosine can apply to three nucleotide contexts: CpG, CpHpG and CpHpH. The equilibrium of the methylation pattern is obtained through the interaction of de novo methylation processes, maintenance of methylation and demethylation. Recently, the methylation of adenine in the N6 position (N6-mA) has also been demonstrated in the genome of *Arabidopsis thaliana*. It has been shown that this modification is essential for plant development and occurs in actively expressed genes [26]. DNA methylation plays an important role in many processes in plants, such as vegetative development, reproduction, fertilization or gametogenesis. Additionally, DNA methylation specifically interacts with other epigenetic mechanisms: posttranslational histone modifications and processes related to non-coding RNA molecules. Due to the constant need to adapt to changing environmental conditions, the “plasticity” of the plant epigenome enables phenotypic traits to be adjusted by specifically regulating gene expression [6].

The use of short oligodeoxynucleotides (ODNs) proved to be a precise method of inducing changes at the epigenomic level. The ODN technology is based on the introduction of short oligodeoxynucleotides of size 12 to 25 nucleotides into cells, which are complementary to the corresponding regions of the target gene. The technology has proven to be an effective method of studying gene function and transformation in animal cells. The method has gained popularity due to the fact that it may impact the modulation of the gene expression of a particular gene and does not lead to modification of the genomic sequence. Currently, ODN technology is optimized to be used in plant research. The favorable property of this method for plant research is the possibility of direct introduction of ODNs to plant cells and limitation of the pleiotropic effects. Although the mechanism of ODN action has not been fully explained yet, it was suggested that the following processes are involved: RNA interference, RNA activation and RNA-dependent DNA methylation [8,9,27]. The majority of reports indicate that the silencing of gene expression is probably the result of the degradation of RNA:DNA duplexes by RNase H [28]. The formation of a triple complex consisting of a DNA double helix and ODN sequence is also not excluded; hence the ability to form a transcript may be limited [29].

The main purpose of our research was to induce significant changes in the transcript level of the CHS gene, which would be maintained after cell divisions. Previously [9], we indicated that the ODN technology is effective in modulation of CHS gene expression and DNA methylation. The changes triggered by the new technology were similar to those observed initially in genetically modified flax (GM-CHS) [9].

Modified ODNs proved to be effective in prolonging the time of maintaining induced changes in methylation of cytosines in the flax genome. Among the two used modifications, methylation of cytosines present in the ODN sequence proved to be more effective than the unmodified version. Analysis of subsequent generations of plants treated with ODNs confirmed the possibility of maintaining induced epigenetic changes during cell division [5]. Due to the declining number of F2 generation plants that maintained the changes induced by unmodified ODNs, F1 generation plants treated with modified oligonucleotides were obtained. The changes induced through ODN6 were maintained in the F1 and F2 generations with the lowest percentage of maintenance. In plants growing in vitro after treatment with ODN6, the tendency of primarily induced changes remains despite subcultures (Appendix A), which confirms the transmission of epigenetic changes during vegetative reproduction [30]. However, at the same time, we cannot exclude that the elevated level of genomic methylation, and hence the reduced expression of the CHS gene, might be triggered by repeated in vitro sub-culturing of plants [31]. More importantly, the modification of ODN 6 through cytosines’ methylation led to improvement of the perpetuation of changes in subsequent generations.

Analysis of methylation of CCGG motifs in the F1 generation of plants treated with ODNs showed the maintenance of the methylation pattern in two of the three analyzed motifs. Instability of methylation status observed in motif +1219/+775 concerned not only plants treated with ODNs, but also control plants. Thus, we presume that particular CCGG motifs may be more susceptible to methylation/demethylation driven by environmental factors, just as the cytosines localized in the transcription factor binding region can be more susceptible to methylation [32]. Despite the fact that in the F1 plants treated with ODN6, the repression of CHS gene was not observed, the methylation pattern of these plants remained after cell divisions. The possible reason for not keeping the CHS expression decreased after the action of ODN6 might be the compensation for effects driven by environmental conditions, and hence increased expression of the CHS gene was observed.

In order to confirm the specific nature of the ODN action, we also determined whether modulation of CHS gene expression displays any changes to the transcript level of the other genes encoding enzymes related to the phenylpropanoid pathway (Appendix A). In plants treated with ODN1, there were no significant changes in the expression of analyzed genes other than *CHS* involved in the synthesis of phenylpropanoids. Only *PAL* and *HCT* showed a minor reduction in gene expression. In the *Arabidopsis thaliana* model plant, HCT gene repression led to a decrease in lignin levels, which is associated with increased CHS gene activity and redirection of metabolic changes to the flavonoid compound synthesis pathway [33]. However, in the flax with CHS gene repression, decreased levels of lignin and HCT gene repression were observed, which did not affect the flavonoid content [19]. Regarding the flax treated with oligodeoxynucleotides modified by methylation, a positive relation was also observed between expression of genes CHS and HCT, where an elevated transcript level of CHS gene was accompanied by increased HCT gene expression. Despite the prolonged maintenance of changes in DNA methylation induced by ODN with thiophosphate bonding, pleiotropic effects concerning the genes related to the synthesis of flavonoids and lignin were observed. Analyzed genes displayed elevated transcript levels. According to a research by [19], enzymes involved in the biosynthesis of flavonoid compounds and lignin occur in the form of multienzyme complexes in which the main regulating component is possibly CHS [19]. Therefore, modulation of CHS gene expression may lead to control of the expression of other genes encoding the components of these complexes. Moreover, in potato transgenic plants overexpressing the CHS gene, effects on other metabolic pathways were also observed, which may be caused by competition for substrates or the effect of the synthesized flavonoids on plant metabolism [34]. However, in plants treated with ODN1 met, no changes in the expression of other genes encoding enzymes involved in the synthesis of phenylpropanoids (Appendix A) were observed, which may suggest that this modification not only stabilizes the modulation of target gene expression, but also increases the specificity of ODN. The group of Khorova have proven that methylation of oligonucleotides increases nucleic acid resistance to nuclease degradation and increases binding strength when interacting with the complementary sequence [35].

DNA methylation is known to cooperate with the positioning of nucleosomes to regulate the structure of chromatin, which leads to an appropriate modulation of gene expression [36]. We have already presented the analysis of the nucleosome energy diagram, which showed that methylation of variable motifs in the CpG island region results in the formation of less stable nucleosomes with higher energy and their shift towards the 3′ end of the DNA [9]. In the vicinity of one of the methylated motifs, sites recognized by AatII and PvuI endonucleases were identified; hence experimental verification of the is silico analysis was performed. In the chromatin structure, DNA regions bound to the histone proteins are less available for endonucleases [37]. Thus, DNA present in nucleosomes is protected from degradation by restriction enzymes. Analysis of chromatin digestion and DNA showed that there is probably a change in the positioning of nucleosomes as a result of the increase in the level of methylation in CCGG +1273 sites.

The present results confirmed that ODN technology induces changes in the plant genome that can be inherited. These data contribute to the statement that the use of natural genetic repertoire through epigenetic variability can be a valuable alternative to improve crop quality.

## 4. Materials and Methods

### 4.1. Designing Short Oligodeoxynucleotide Sequences (ODN) Homologous to CHS Gene Regions

On the basis of our previous work [9], for this study, three ODNs homologues to the particular CHS gene regions were selected: ODN1 (5′UTR), ODN6 (intron) and ODN11 (3′UTR). The ODN sequences were designed using Mfold software (version 3.2, Genetics Computer Group, Madison, WI, USA) in the antisense orientation and correspond simultaneously to two CHS gene isoforms: LuCHS6 (Phytozome data base: Lus10033717) and LuCHS7 (Phytozome database: Lus10031622). Detailed information about used ODN sequences were reported by [9] (Table 2, Appendix A). In order to enhance the resistance of the sequences to the degradation by nucleases, the sequences of the three mentioned oligonucleotides were modified by methylation (all cytosines occurred in the sequences) and thiophosphate modification (at each phosphodiester bond). The oligodeoxynucleotides were synthesized by Genomed S.A. (Warszawa, Poland).

### 4.2. Plant Material

Flax seeds *(Linum usitatissimum* L., cv. Linola) were obtained from the Flax and Hemp Collection of the Institute of Natural Fibers (Poznań, Poland).

#### Flax Growing Conditions for ODN Technology

To investigate whether modified ODNs are more effective in the stability of the induced epigenetic modulation, flax plants were cultured in in vitro conditions in the phytotron at 16 h light (22 °C), 8 h darkness (16 °C). The seeds were first sterilized for 15 min with 50% PPM—a broad-spectrum biocide/fungicide for plant tissue culture (Plant Preservative Mixture; Plant Cell Technology, Washington, DC, USA) and then germinated on Murashige and Skoog (MS) basal medium (Sigma-Aldrich, St. Louis, MO, USA), supplemented with 2% sucrose, pH = 5.8, solidified with 0.8% agar on Petri dishes. Mature flax plants were cultured on the MS basal medium, supplemented with 2% sucrose, pH = 5.8, solidified with 0.8% agar in sterile glass jars. In order to minimize plant infection by pathogens, the medium was complemented with 0.1% PPM™ (Plant Preservative Mixture) Plant Cell Technology (Washington, USA).

In order to obtain seeds for raising the next generation of plants in the field, *Linola* flax seeds were germinated in the soil in the phytotron conditions in 16 h light (22 °C), 8 h darkness (16 °C), before they were treated with ODNs.

### 4.3. Identification of cDNA Sequences

The isolated mRNA from Linola flax was submitted to sequencing. Necessary sample preparations, sequencing and data processing were performed by Genomed S.A. (Warszawa, Poland). Data from three biological replicates were analyzed.

### 4.4. Flax Treatment with ODNs by Infiltration

Four-week-old Linola flax cultured in in vitro conditions were used for the treatment with modified ODNs. The plants were cut above their roots and submerged in water solution of the particular ODN in the 10 µM concentration. The incubation was carried out in the vacuum chamber for 20 min. After the treatment, the plants were put into an MS medium. The material for the analysis was harvested 10 days after infiltration. The harvesting time was indicated by previous experiments [8] and enabled determination of the ODNs’ activity in comparison to the unmodified sequences.

For the field cultivation, whole two-week-old seedlings germinated in the soil were treated with the ODNs by the forced osmosis method. It was described in [38] that the sugar starvation method of introducing ODNs is effective in the induction of changes in DNA methylation. First, the plants were transferred from the soil into the water and kept in the darkness in the phytotron chamber in order to deplete endogenous sucrose. Then seedlings were put in the water solution of 100 mM sucrose in the presence of 10 µM antisense ODNs. After 24 h of incubation in the darkness with particular oligodeoxynucleotides, plants were put into the soil and cultivated in the experimentation field. The matured seed capsules were harvested approximately four months later. Collected seeds were stored in dry, cool conditions. In order to determine the stability of induced modifications in the subsequent generations, some of the seeds were sterilized and cultured in in vitro conditions.

### 4.5. Flax cultivation in the Field

In general, the growing season for flax in a moderate climate is between April and July. Seeds collected from plants treated with ODNs were sown in the subsequent season to obtain an F1 generation of plants. For an F2 generation, plants were cultivated in the analogous manner.

### 4.6. Gene Expression Analysis

The expression of investigated genes was analyzed via real-time PCR. The total RNA was isolated from freeze-ground green tissue of the transgenic plants. Isolation was performed using the Trizol method (Invitrogen, Carlsbad, CA, USA), following the protocol of the manufacturer. The isolated RNA was deprived of DNA contamination by DNase I (Invitrogen, Carlsbad, USA). The purified RNA was used as a template for cDNA synthesis using reverse transcriptase—High Capacity cDNA Reverse Transcription Kit (Applied Biosystems, Foster City, CA, USA).

Real-time PCR reactions were performed using a DyNAmo SYBR Green qPCR kit (Thermo Scientific, Waltham, MA, USA). Primers used for the reaction are presented in the Appendix A. The used system was the Applied Biosystems StepOnePlus Real-Time PCR System (Applied Biosystems, Foster City, USA). The reaction was conducted according to the protocol provided by the manufacturer. Each reaction was performed in three repeats. As the reference, the actin gene was used. The results were presented as the relative quantification (RQ) to the reference gene.

### 4.7. DNA Methylation Patterns in Specific Regions of Chalcone Synthase Gene

The methylation patterns of the chalcone synthase gene sequence were established in the control and ODN-treated plants. The DNA was incubated with restriction enzymes MspI and HpaII for at least 3 h (New England Biolabs, Ipswich, MA, USA); it differs in sensitivity to cytosine methylation. The genomic DNA digested by the restriction enzymes and undigested DNA were used as a templates for the real-time PCR reaction. The reaction was performed similar to the gene expression analysis. The primers for the reaction (Appendix A) were designed for specific sites of methylation predicted by the NEB cutter V2.0. In the chalcone synthase gene, six CCGG islands were analyzed. The sites of analyzed -CCGG- motifs were indicated by their positions towards the ATG site (+1) as follows: 5′UTR (site −232), non-coding region (+217) and coding region (+996, +1219, +1273, +1606).

The quantity of DNA measured by the real-time PCR was calculated in order to estimate the methylation of cytosines. The “CCGG” non-methylated DNA was calculated as the difference between undigested DNA and DNA incubated with HpaII. The single methylated cytosine “CCmGG” was estimated as the difference between samples digested by HpaII and digested by MspI. The level of the “CmCmGG” was equal to the DNA incubated with MspI. The values were presented as a percentage referring to the undigested DNA, set as 100%.

### 4.8. Relative Accessibility of Chromatin and DNA

The chromatin was isolated from 1 g of freeze-ground green tissue by ChromaFlash Plant Chromatin Extraction Kit (EpiGentek, Farmingdale, NY, USA) according to a protocol provided by the manufacturer. The sonication stage was set for the QSonica 700 (QSonica, Newtown, CT, USA) instrument and performed with the following parameters: Amplitude–25%, Process time–3 min 20 s, Pulse ON–20 s, Pulse OFF–30 s. Isolated chromatin was divided into portions (approximately 300–400 ng per aliquot), snap-frozen in liquid nitrogen, and stored at −80 °C.

Digestion of chromatin and DNA was performed. First, the chromatin was digested with the following FastDigest enzymes: AatII and PvuI (Thermo Scientific, Waltham, MA, USA). The restriction enzymes were defined by in silico analysis. Sites of digestion for AatII and PvuI are located in the nucleosomes, where DNA wraps around nucleosomes. After digestion with restriction enzymes (15 min, 37 °C) and enzymes’ inactivation (AatII, PvuI –80 °C, 5 min), the samples were incubated with Proteinase K (Qiagen, Hilden, Germany) and RNase A (Qiagen, Hilden, Germany) at 37 °C for 1 h. Simultaneously, chromatin samples were incubated first with Proteinase K and RNase A, in order to release the DNA from the nucleosome. The DNA was digested with restriction enzymes in an analogous manner as chromatin was cut. Undigested control samples were also prepared. Total DNA was purified from each sample with phenol/chloroform/isoamyl alcohol (PCI-25:24:1, *v*/*v*), mixed vigorously and centrifuged (10,000× *g*, for 5 min, at room temperature (RT)). The upper phase was transferred into a new tube and DNA was precipitated by adding 0.5 mL of cold isopropanol. In order to facilitate precipitation, 2 µL (20 µg) of linear polyacrylamide was added (5 mg of acrylamide was dissolved in 200 µL of water, 1 µL of 10% APS and 1 µL of TEMED were added, left to polymerase overnight at RT, 2.5 volumes of ethanol were added, centrifuged at 12,000× *g*, 5 min, dried and re-suspended in 500 µL of sterile water). After 10 min of incubation on ice, the samples were centrifuged 12,000× *g*, for 15 min, at 4°C. The DNA pellet was washed with 0.5 mL of cold 70% (*v*/*v*) ethanol and centrifuged at 12,000× *g* for 10 min at 4 °C. The dry pellet was re-suspended in 40 µL of sterile water.

The relative accessibility of chromatin/DNA was assessed by the real-time PCR method. During the reaction, the region surrounding motif CCGG located at +1273 was amplified. The selection of this region was preferred due to the methylation status of the particular motif and the location of the sites of cutting by AatII and PvuI restriction enzymes. The real-time PCR reaction was performed similarly to the gene expression analysis. The quantity of measured DNA was calculated in order to estimate the relative accessibility of chromatin/DNA to the restriction cut. The actin gene was used as the reference. The results were presented as the relative chromatin/DNA accessibility in comparison to the undigested control, set as 1.

### 4.9. In Silico Analysis

#### Chromosome Localization of CHS Genes from Flax

The genomic sequence of each CHS gene was aligned in BLASTn (Available online: https://blast.ncbi.nlm.nih.gov/Blast.cgi) to whole genome sequencing of *Linum usitatissimum* [39]. The localization (range) of exons in the chromosome was identified for the particular gene.

### 4.10. Statistical Analysis

Presented experiments were performed at least three times, independently (three technical repeats of each of three biological samples). Data represent the mean value ± standard deviation (SD) from at least three independent experiments. The significance of the differences between each mean and control was determined by Student’s t-test. An asterisk above each bar indicates *p* < 0.05. The statistical analysis was performed using STATISTICA (ver. 12, StatSoft Inc.—Dell, Round Rock, TX, USA).

## 5. Conclusions

In this article, we presented the latest research regarding ODN technology in flax, considering two genes: *LuCHS6* and *LuCHS7*. The main purpose was to demonstrate the potential maintenance of induced epigenetic changes in CHS gene expression and methylation via short oligodeoxynucleotides complementary to the gene of interest. Induced changes in the methylation status (5-methylcytosine) and CHS gene expression can be perpetuated in the next generation. Modified ODNs (especially, by methylation) were assumed to be successful in prolonging the ODN-triggered effect. Although the mechanism of ODNs’ action and heredity of ODN-induced changes are not fully explained yet, the processes involving the sRNAs (small non-coding RNAs) and RdDM (RNA-directed DNA methylation) along with the rearrangement of nucleosome positioning are suspected to play a substantial role in defining the epigenome [8,9]. Thus, we suggested the great potential of ODN technology in plant research as a significant tool not only for genetic and epigenetic studies, but also crop improvement.

## Figures and Tables

**Figure 1 ijms-20-03983-f001:**
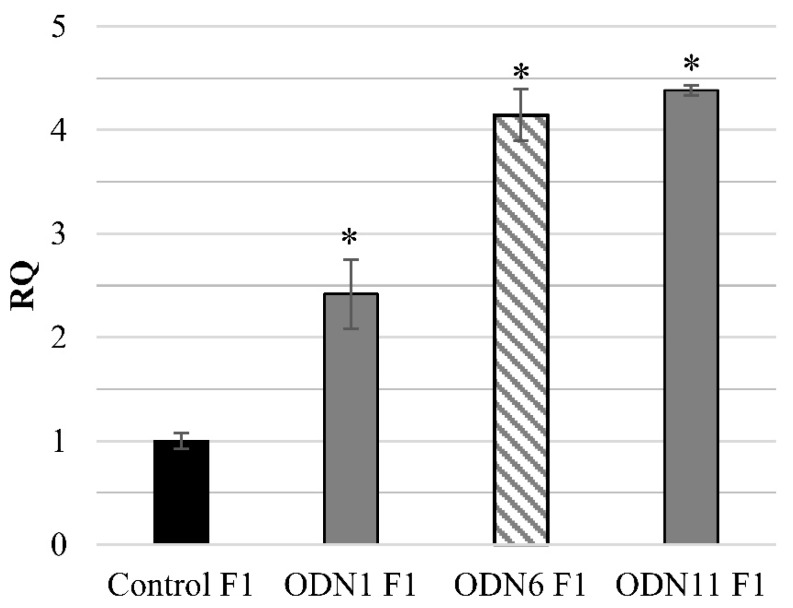
CHS gene expression in the F1 generation of plants that maintained modifications induced by short oligodeoxynucleotides (ODNs), cultivated in the experimental field. The seeds obtained from F0 plants were sown and plants were cultivated in the field. The total expression of both CHS genes, *LuCHS6* and *LuCHS7*, was determined by the real-time PCR reaction. The values are referred to the reference gene expression actin. The relative quantity (RQ) presents the transcript level in comparison to the control (set as 1, black). Plants treated with ODN6 met did not maintain in the F1 generation the initially observed level of CHS gene expression (bar presented in diagonal stripes). Data represent the mean value ± SD from at least three repeats of the experiment. The significance of the differences between each mean and control was determined by Student’s *t*-test. Asterisk indicates * *p* < 0.05.

**Figure 2 ijms-20-03983-f002:**
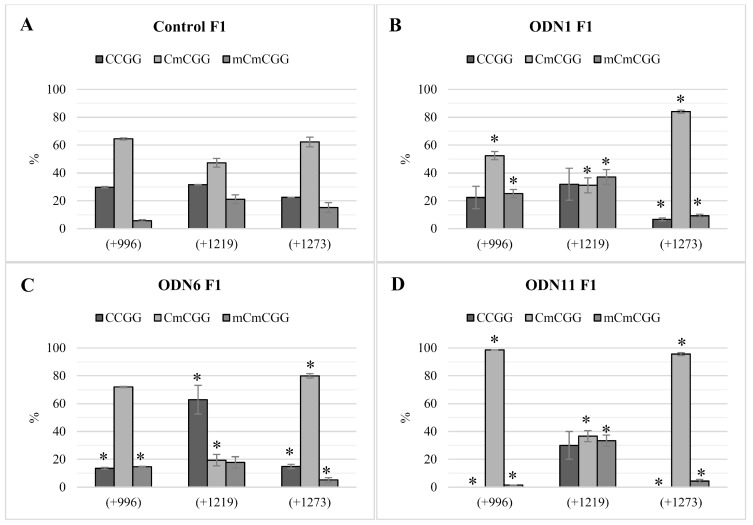
Methylation in the variable CCGG motifs identified in the CHS gene sequence in the F1 generation of control plants (**A**) and individuals that maintained modifications induced by ODNs (**B**–**D**), cultivated in the experimental field. The seeds obtained from F0 plants were sown and plants were cultivated in the field. The figure presents the percentage of cytosine methylation in the variable CCGG sites (+996, +1219, +1273) located in the coding region of the CHS gene sequences. The genomic DNA was digested by restriction enzymes HpaII and MspI. The amount of non-digested DNA was determined by real-time PCR. The percentage of a particular modification is presented for studied plants and control: CCGG—lack of methylation (dark grey), CCmGG—methylation of internal cytosine (light grey) and CmCmGG—methylation of both cytosines (medium grey). The site positions were presented according to the *LuCHS6* sequence, due to the presence of all CCGG sites. Data represent the mean value ± SD from at least three independent + experimental repeats. The significance of the differences between each mean and control was determined by Student’s *t*-test. Asterisk indicates * *p* < 0.05.

**Figure 3 ijms-20-03983-f003:**
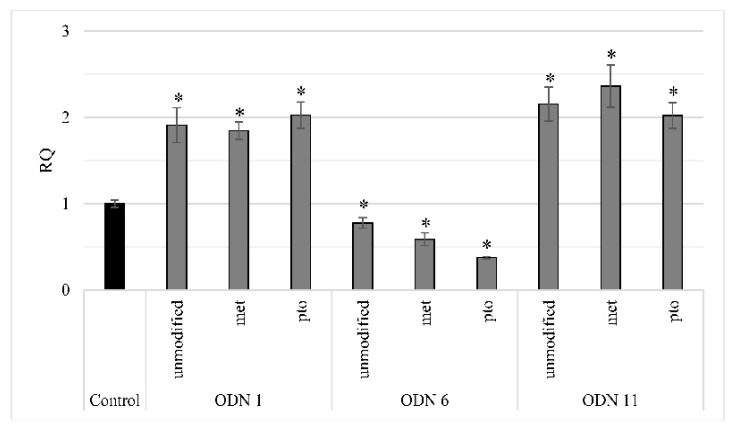
CHS gene expression in in vitro plants treated with modified ODNs after 10 days of incubation. The 4-week-old plants cultured in vitro were incubated with particular sequences of previously analyzed ODN 1, 6 and 11: unmodified ODN, ODN met (sequence with methylated cytosines) and ODN pto (sequence with thiophosphate bonds). The total expression of the two CHS genes *LuCHS6* and *LuCHS7* was determined by the real-time PCR reaction in plants harvested 10 days after incubation. The values are referred to the reference gene expression actin. The relative quantity (RQ) presents the transcript level in comparison to the control (set as 1, black). Data represent the mean value ± SD from at least three independent experiments. The significance of the differences between each mean and control was determined by Student’s *t*-test. Asterisk indicates * *p* < 0.05.

**Figure 4 ijms-20-03983-f004:**
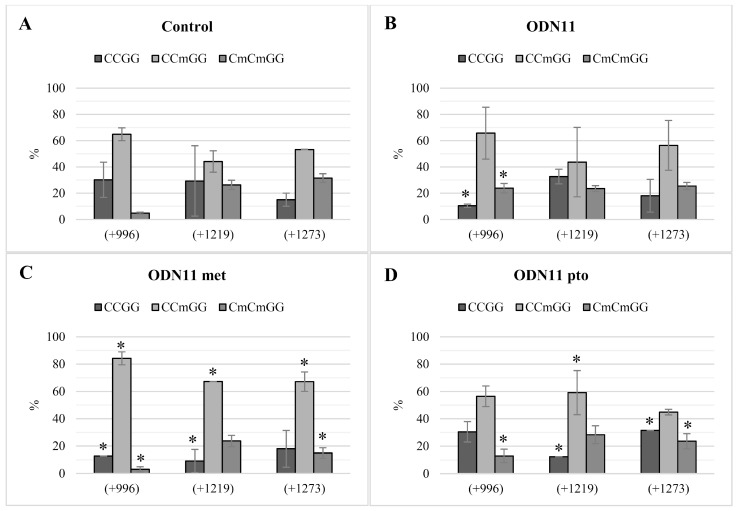
Methylation profiles for variable CCGG motifs in control plants (**A**). Plants treated with unmodified ODN11 (**B**) and modified ODN11 met (**C**) and ODN11 pto (**D**), 10 days after the treatment. The figure presents the percentage of the cytosine methylation in the variable CCGG sites (+996, +1219, +1273) located in the coding region of the CHS gene sequence. The genomic DNA was digested by restriction enzymes HpaII and MspI. The amount of non-digested DNA was determined by real-time PCR. The percentage of particular modification was presented for studied plants and control: CCGG—lack of methylation (dark grey), CCmGG—methylation of internal cytosine (light grey) and CmCmGG—methylation of both cytosines (medium grey). The site positions were presented according to the *LuCHS6* sequence, due to the presence of all CCGG sites. Data represent the mean value ± SD from at least three independent experiments. The significance of the differences between each mean and control was determined by Student’s *t*-test. Asterisk indicates * *p* < 0.05.

**Figure 5 ijms-20-03983-f005:**
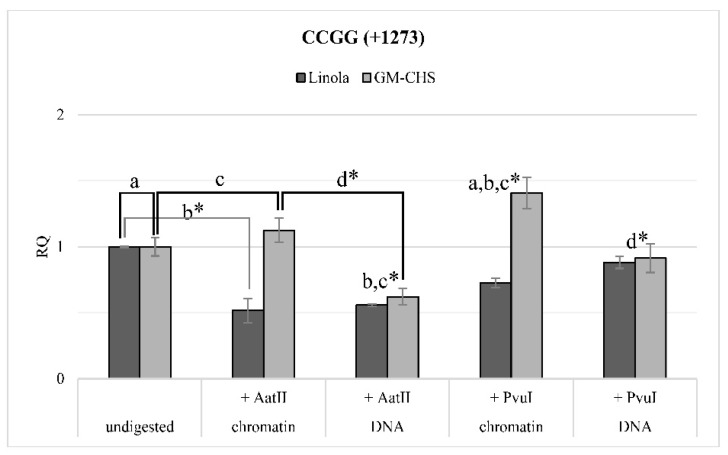
Chromatin/DNA accessibility assay. Chromatin from non-transgenic (Linola) and GM-CHS flax (W92.40) was isolated using a dedicated technical kit. The chromatin and chromatin-derived DNA (primarily treated with proteinase) were incubated with restriction enzymes AatII and PvuI (particular sites for digestion were localized in the CHS gene). After being cut, the DNA was purified and the effect of DNA digestion was assessed by real-time PCR at the methylation CCGG site (+1273). The values are referred to the reference gene expression actin. The relative quantity (RQ) presents the transcript level in comparison to the samples not incubated with the restriction enzymes (undigested, set as 1 for each line). Data represent the mean value ± SD from at least three independent experiments. The significance of the differences between the samples was determined by Student’s *t*-test: a—the reference of the methylated motif to unmethylated, b—in reference to the undigested sample for unmethylated motif, c—in reference to the undigested sample for methylated motif, d—the reference of the DNA to chromatin, for the appropriate restriction enzyme. Asterisk indicates * *p* < 0.05.

**Table 1 ijms-20-03983-t001:** Genes encoding CHS in flax.

Gene	Gene ID (Phytozome)	Gene Accession No (GenBank)	Chromosomal Localization
*LuCHS3* ^a^	Lus10041508 ^a^	AFSQ01010505 ^b^	CP027628.1 (Lu4) Range 1: 14557172 to 14557366 Range 2: 14558491 to 14559582
*LuCHS4* ^a^	Lus10023670 ^a^	AFSQ01007984 ^b^	CP027629.1 (Lu5) Range 1: 8911370 to 8911554 Range 2: 8911645 to 8912061 Range 3: 8912194 to 8912636
*LuCHS5* ^a^	Lus10011746 ^a^	AFSQ01007986 ^b^	CP027629.1 (Lu5) Range 1: 8892102 to 8892257 Range 2: 8893137 to 8894143
*LuCHS6* ^a^	Lus10033717 ^a^	AFSQ01005163 ^b^	CP027630.1 (Lu6) Range 1: 1398135 to 1398320 Range 2: 1396599 to 1397611
*LuCHS7* ^a^	Lus10031622 ^a^	AFSQ01023955 ^b^	CP027622.1 (Lu12) Range 1: 19609154 to 19609339 Range 2: 19609419 to 19610428
*LuCHS10*	-	AFSQ01010603 ^b^	CP027619.1 (Lu1) Range 1: 5753813 to 5754010 Range 2: 5753070 to 5752694
*LuCHS11*	-	AFSQ01012744 ^c^	CP027632.1 (Lu8) Range 1: 14778517 to 14778701 Range 2: 14777435 to 14778426

^a^—indicates the CHS genes (sequences and names) published by [20]; ^b^—indicates the CHS genes (sequences and names) published by [21]; ^c^—recently identified CHS gene.

**Table 2 ijms-20-03983-t002:** Maintenance of gene expression modulation of CHS gene in F1, F2 and F3 generations of plants treated with short oligodeoxynucleotides (ODNs).

Generation	Type of Plant	Percentage of Modulation Maintenance (%)
F1	ODN1	71
	ODN6	5
	ODN11	86
F2	ODN1	33
	ODN6	5
	ODN11	75
F3	ODN1	38
	ODN6	61
	ODN11	6

**Table 3 ijms-20-03983-t003:** Maintenance of gene expression modulation of CHS gene in F1 and F2 generations of plants treated with modified ODNs (met and pto).

Generation	Type of Plant	Percentage of Modulation Maintenance (%)
F1	ODN1 met	80
	ODN6 met	50
	ODN11 met	80
	ODN11 pto	100
F2	ODN6 met	50
	ODN11 met	38
	ODN11 pto	47

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
