# Peer review of "Transgenerational Perpetuation of CHS Gene Expression and DNA Methylation Status Induced by Short Oligodeoxynucleotides in Flax (Linum usitatissimum)"

_ijms, 2019, doi:10.3390/ijms20163983_

Round 1
Reviewer 1 Report
The article submitted by Magdalena Zuk and co-workers, entitled “Transgenerational Perpetuation of the CHS Gene Expression and DNAMethylation Status Induced by Short Oligodeoxynucleotides presents a set of very interesting data introducing effect of short oligodeoxynucleotides (ODNs) on chalcone synthase, an enzyme involved in biosynthesis of flavonoids, on selected signatures of epigenome, i.e. DNA methylation level, chromatin rearrangement of the Linum unisatissimum experimental model.
The presented data well confirm and develop earlier work of the group.
- The introduction is coherent and well brings into the topic of the paper.
- The methods for validation the hypothesis are properly chosen and described in sufficient detail.
- The studies performed, are properly designed and controlled, replicated suitable.
- The presented data are of high quality.
- Statistical analysis of data is performed properly.
- The conclusions reached, are consistent with the presented data.
Comments to be addressed:
(1) The abstract of the article should be modified. The information about previous results of the group are included in the introduction and should be removed from the current version of the abstract (sentence no. 2), as well as future plans (sentence no. 5). The authors should focus on the presented article and give wider feedback about the obtained results. Also the experimental model should be better indicated, presentation of the Latin name of the plant should be added.
(2) Some minor typing errors and grammar mistakes should be removed, e.g. line 19 “we focused” or we will focused; line 118-119 the nomenclature ... will be followed – or was – or “in the presented paper the nomenclature will be followed”? line 159, a space between ‘dot’ and ‘The seeds’ is missing, as in line 161 - 6weeks/6 weeks....
(3) Figure 2/Figure 4. I suggest to remove the values that are presented under the graphs, all is clearly visible and double presentation of the same data is not necessary.
I do recommend the publication of the paper after including the indicated comments.
Author Response
The article submitted by Magdalena Zuk and co-workers, entitled “Transgenerational Perpetuation of the CHS Gene Expression and DNA Methylation Status Induced by Short Oligodeoxynucleotides presents a set of very interesting data introducing effect of short oligodeoxynucleotides (ODNs) on chalcone synthase, an enzyme involved in biosynthesis of flavonoids, on selected signatures of epigenome, i.e. DNA methylation level, chromatin rearrangement of the Linum unisatissimum experimental model.
The presented data well confirm and develop earlier work of the group.
- The introduction is coherent and well brings into the topic of the paper.
- The methods for validation the hypothesis are properly chosen and described in sufficient detail.
- The studies performed, are properly designed and controlled, replicated suitable.
- The presented data are of high quality.
- Statistical analysis of data is performed properly.
- The conclusions reached, are consistent with the presented data.
We are very happy about the positive opinion about our work. Thank you very much.
Comments to be addressed:
(1) The abstract of the article should be modified. The information about previous results of the group are included in the introduction and should be removed from the current version of the abstract (sentence no. 2), as well as future plans (sentence no. 5). The authors should focus on the presented article and give wider feedback about the obtained results. Also the experimental model should be better indicated, presentation of the Latin name of the plant should be added.
We removed from the abstract information about earlier our results (2nd sentence of the abstract). However, sentence 5 refers to the results presented in the current work. The misunderstanding resulted most probably from the unfortunate expression we used. Now it has been corrected.
The Latin name of flax-our research model has been added.
(2) Some minor typing errors and grammar mistakes should be removed, e.g. line 19 “we focused” or we will focused; line 118-119 the nomenclature ... will be followed – or was – or “in the presented paper the nomenclature will be followed”? line 159, a space between ‘dot’ and ‘The seeds’ is missing, as in line 161 - 6weeks/6 weeks....
All grammatical and punctuation errors mentioned above have been corrected. In addition, the article has been improved by the company professionally correcting English correction in scientific articles which resulted in the correction of other errors as well. In order to distinguish the language correction from the Reviewers suggestions, each correction made by Authors (according to Reviewers suggestion) was described in this letter, accordingly.
(3) Figure 2/Figure 4. I suggest to remove the values that are presented under the graphs, all is clearly visible and double presentation of the same data is not necessary.
The values above the graphs have been removed, as suggested by the reviewer.
I do recommend the publication of the paper after including the indicated comments.
Thank you again for the review.
Yours faithfully
Authors
Reviewer 2 Report
Dear Authors,
I recommened the manuscript of Dzialo et al. „Transgenerational Perpetuation of the CHS Gene
Expression and DNA Methylation Status Induced by Short Oligodeoxynucleotides” for considering the acceptance after major revision.
The main purpose of this manuscript was the confirmation of the maintenance of induced epigenetic changes in CHS gene expression and methylation via short oligodeoxynucleotides complementary to the gene of interest (LuCHS6 and LuCHS7, ODN technology) in flax. The Authors pointed out that the induced changes in the methylation status can be perpetuated in the next generations. They suggest that the ODN technology is a well acceptable technology not only for genetic and epigenetic studies, but also crop improvement.
The project itself is good, acceptable and shall be suitable for publication if the Authors shall be able to correct the below listed problems.
General comments:
My feeling that this MS was already rejected in an other journal and after that it was sent to IJMS without any correction. This is again a MS which was submitted to the IJMS without taking into account the specific requirements of this journal, especially as for References. This part of this MS is completely wrong, does not follow the Instructions for the Authors section of IJMS, this shall have to completely rewrite. Secondly, the paper requires an extensive editing of English language and style. Third problem is the the description of Mat&Met, the Authors shall have to correct a lot of defects here but there are other problems with the MS which are listed below.
Minor comments:
Introduction:
line 36: the right sentence: „methylation status enable to control the chromatin condensation”
line 44: erosion instead of erasion
line 77: „due to:” delete (:)
line 84: instead of Eom and Hyun 2016, please write: according to [Reference with number]
line 102: citation is not good in the Table 1 Figure legends what you should create, e.g. putting the Table1. Genes encoding CHS in flax sentence below the table.
And instead of Eom and Hyun 2016, please write: published by [Reference with number]. The same for Boba et al. 2017 like published by [Reference with number].
line 105: right sentence: since the beginning moment of the incubation
Results:
line120: „the numbers used by the Eom et al. (Eom et al, 120 2016),” – this is duplication as the reference and you shall have to use the „used by [Reference number]” worlds.
line 122: „The inclusively level of the mRNA derived from…”
line 131: „ The part of seeds obtained” - it is better to write : A part of seeds obtained…
line 139: The „Seeds collected from the F1 generation” is better
line 178: „Asterisk indicates P < 0. 05” is better
line 180: „For the presented in the Figure 1 plants” is wrong. For the plants presented in Figure 1 is better
lines 183-188: this section shall have to rewrite because simply ununderstandable
line 190: after 48 h since the moment of incubation”. Better to write: after 48 h after incubation
line 193: „which corresponded with primarily induced by ODNs modifications” May be it is better:
which correlates with primarily induced ODNs modifications
line 199: ” Data constitute the mean value ± SD from at least three repeats of the experiment.” Instead: Data constitute the mean value ± SD from at least three independent +experimental repeats.
line 204: „the modification of the nucleotides included”. Replace with: two modification of the nucleotides included..
line 208: 2.3.1. „CHS gene expression after 10 days since the incubation with ODNs”. Replace with: 2.3.1. CHS gene expression studies 10 days after ODNs incubation.
line 212: Right: 10 days form the beginning of incubationsince the moment of incubation …..
line 2013: after ODN1 and ODN11 treatments
line 218: In Figure 3 legends: modified ODNs after 10 days of incubation
line 222: Correct for: in plants harvested after 10 days after incubation
line 255: the specification of CCGG sites should have already introduce in the Figure 2. Please, write for Fig2 legend this also
line 251: Figure 4. The methylation profiles for variable CCGG motifs in control plants (A). Plants treated with
unmodified ODN11 (B) and modified ODN11 met (C) and ODN11 pto (D), after 10 days after the beginning of the treatment. This better a bit….
line 259: „Data constitute represent the mean value ± SD from”
line 261: better: Asterisk indicates P < 0.05.
line 263: Better: flax cultivated in vitro.
line 264: The A part of the seeds obtained seeds were partially sterilized and introduced to the in vitro culture.
line 269: „Regarding ODN6, the modification of oligonucleotide sequences via methylation allowed to select one individual with the repressed level of the CHS expression.” From what Figure did you get this conclusion? The number of plants is not represented for ODN6.
line
line 275: „In the comparison to the F1 generation the number of analyzed plants were higher.” Again, where the plant numbers are indicated in the text?
line 278: Table 3. legend shall have to put under the Table.
line 282: The chromatin isolated from in vitro cultured plants….. is better
line 291: „CCGG motif increased level of..” is better
line 306: Better sentence: Asterisk indicates P < 0.05.
Discussion:
line 311: instead permanent use stable transformant
line 319: including TALENs, ZFNs and CRISPR/Cas9 [reference needed].
line 332: „pattern is obtained through the interaction of: „ (:) is not necessary to put
line 336:” such as:” (:) is not necessary to put
line 344: „size 12 to 25 nucleotides into cells” is better
line 351: „the following processes may be are involved:”
line 357: „The main purpose of our research was to induce the significant changes in the transcript level of the CHS gene,”
line 358: „Previously in Dzialo et al, 2017”. Right citation: Previously in []…..
lines 366-368: These sentences are ununderstable ones. Please, revise it.
line 368-369. „The changes induced through ODN6 were maintained in the F1 and F2 generations with the lowest percentage of maintenance.” Do you have any explanation for this?
line 371: „despite passages „ Please, always use subculture instead of passage (not only here but everywhere in the MS)
lines 396-400: this is ununderstandable, please revise this sentences.
line 401: „According to the research of Żuk et al. (2016)” Right citation: According to the research of []
lines 408- 409: „enzymes involved in the synthesis of phenylpropanoids were observed (Supplemental Figure 2) which”….,
Mat&Meth:
line 430: „On the basis of our previous work Dzialo, et al. 2017,” right citation is : On the basis of our previous work []….
line 432: Mfold software (Genetics Computer Group, city name is missing, USA)
line 435: can be found in []
line 439: Genomed S.A. (city name is missing, Poland).
line 442: Institute of Natural Fibers (city name is missing, Poland)
line 446: Please, write the sterilization conditions or refer to relevant literatura here.
line 446: „The seeds were germinated on Murashige and Skoog (MS) basal medium (Sigma-Aldrich, city, country names are missing),
line 448: Mature flax plants were cultured on the MS medium, supplemented with 2….
lines 450-451: the medium was complemented with 0,1% PPMTM (Plant Cell Technology, city, country names are missing). You shall have to specify what the PPM is !
line 457 data processing was performed by Genomed S.A. (city name is missing, Poland).
line 462-463: particular ODN at 10μM concentration. The incubation time was 20 minutes in a vacuum chamber...
line 463-464: The material for the analysis was harvested 10 days after infiltration.
line 465: You shall have to put a reference here
line 467: „Sun et al, 2005 has shown” It was described in []…
lines 473-474: The matured seed capsules were harvested after approximately 4 months later. since plant germination.
line 478: In general, the growing season for flax in the moderate climate (Poland) span between April and July in Poland.
lines 485, 486, 488: Trizol method (Invitrogen, Applied Biosystems, so on: city, country names are missing),
line 490: Thermo Scientific, city, country names are missing)
line 490: From Supplemental Table 1 the accession numbers of the CMT1, CMT3, DME, ROS1, DDM1, H3K9 genes are missing. Please, replace it.
line 491: system was Applied Biosystems StepOnePlusTM Real Time PCR System – where from it arised, city, country names are missing
line 497: The DNA was incubated with restriction enzymes MspI and HpaII for at least 3 hours (restriction enzymes MspI and HpaII (New England Biolabs)…..
line 498: New England Biolabs, city, country names are missing
line 502: „In the synthase chalcone gene there” better: In the chalcone synthase gene there
line 514: Chromatin Extraction Kit (EpiGentek, city, country names are missing)
line 515: set for QSonica 700 instrument (city, country names are missing)
line 520: AatII and PvuI (Thermo Scientific, city, country names are missing).
line 524: (Qiagen city, country names are missing).) and RNAse A (Qiagen city, country names are missing)
line 526: The Undigested
line 531: (5mg of acrylamide…
line 534: were centrifuged at 12,000 x g, for 15 min, at 4°C.
line 539: The selection of this region was preferred acknowledged due to the methylation status of the particular motif
line 540-541: The Real Time PCR reaction was performed similarly to the gene expression analysis.
line 549: The localization (range) of exons in chromosome was for identified for the particular gene.
line 564: sRNAs and RdDM specify what they are
line 566: epigenome [] - reference is necessary here.
References:
The whole References shall have to replace with a corrected new one where the papers are cited according to the IJMS basic regulations.
Best regards,
Reviewer
Author Response
Please find the attached file with the answers.

Round 2
Reviewer 2 Report
Dear Authors!
Thank you very much for the carefull corrections. The paper is now acceptable after very minor corrections listed below.
Pleae, correct these sentences:
line 86: the right sentence should be the next:
It was demonstrated that there are 5 CHS genes (Phytozome ID): ) – LuCHS3 (Lus10041508), LuCHS4 (Lus10023670), LuCHS5 (Lus10011746), LuCHS6 (Lus10033717), LuCHS7 (Lus10031622) – and 4 CHS-like genes: (Phytozome ID): ) – LuCHS1 (Lus10039904), LuCHS2 (Lus10042388), LuCHS8 (Lus10026286), LuCHS9 (Lus10002187) – in the genome of Linum 90 usitatissimum L. [20].
line 108: „c -– recently identified CHS genes.” There is only one gene here, please correct for „gene”.
line 468: „Murashige and Skoog (MS) basal medium (St. Louis, Sigma-Aldric)” – the country is still missing here
line 467: „with 50% PPM - Plant Preservative Mixture (Washington, Plant Cell Technology)” – here you should mention what the PPM is, and the the country is still missing here also
line 472: the right snetence should be: with 0,1% PPM™ (Plant Preservative Mixture) Plant Cell Technology, Washington,US).
There are a lot of mistakes yet in the References, please, correct them!
Ref. 2: Robertson, A.; Wolf, D. The role of epigenetics in plant adaptation; 2012; Vol. 4, pp. 4
Title or the Journal name as well as the pp is missing.
Ref. 3. The journal name is: Trend Plant Sci. and the volume and pp are missing!
Ref. 5. no final page is added
Ref. 6. Right one: Cold Spring Harbor Perspectives in Biology 2014,….
Ref. 7. Right one: Trends in Cell Biol. 2014, 24, 100-107.
Ref. 8. Right one: BMC Plant Biol. 2014, 14, 261-?,
Ref.13. Right one: Nutrition & Metabolism 2012, 9, 8-?.
Ref. 19. Right one: Frontiers in Plant Sci. 2016
Ref. 21. Right one: Frontiers in Plant Sci. 2017
Ref. 22. Right one: Theor. Appl. Genet. 2016,…
Ref. 23. Right one: J. Clinical Invest. 2014, 124…
Ref. 26. Right one: Dev. Cell 2018, 45…
Ref.30. Right one: Plant Cell Physiol. 2014,…
Ref .33. Right one: The Plant Cell 2007,..
Ref .37. Right one: Meth. Mol. Biol.
Ref. 38. Right one: Plant J. Cell Mol. Biol. 2005, 44…
Ref. 38. Right one: Plant J. Cell Mol. Biol….
Sincerely yours,
Reviewer 2
Author Response
Thank you very much for the careful corrections. The paper is now acceptable after very minor corrections listed below.
Thank you for the revision.
Please find the explanation for every correction in the description below.
The sentences:
line 86: the right sentence should be the next:
It was demonstrated that there are 5 CHS genes (Phytozome ID): ) – LuCHS3 (Lus10041508), LuCHS4 (Lus10023670), LuCHS5 (Lus10011746), LuCHS6 (Lus10033717), LuCHS7 (Lus10031622) – and 4 CHS-like genes: (Phytozome ID): ) – LuCHS1 (Lus10039904), LuCHS2 (Lus10042388), LuCHS8 (Lus10026286), LuCHS9 (Lus10002187) – in the genome of Linum 90 usitatissimum L. [20].
Corrected according to the Reviewer’s suggestions and the citation in the line 85 was deleted.
line 108: „c -– recently identified CHS genes.” There is only one gene here, please correct for „gene”.
Corrected according to the Reviewer’s suggestions (line 105).
line 468: „Murashige and Skoog (MS) basal medium (St. Louis, Sigma-Aldric)” – the country is still missing here
Corrected to “(Sigma-Aldrich, St. Louis, USA)” (line 455)
line 467: „with 50% PPM - Plant Preservative Mixture (Washington, Plant Cell Technology)” – here you should mention what the PPM is, and the the country is still missing here also
Corrected according to the Reviewer’s suggestions (lines 452-454)
line 472: the right snetence should be: with 0,1% PPM™ (Plant Preservative Mixture) Plant Cell Technology, Washington,US).
Corrected according to the Reviewer’s suggestions (lines 458-459)
References:
All mistakes in the “References” section were corrected according to the Reviewer’s suggestions.
We noticed that the wrong reference was cited in the position no. 3, hence the citation was replaced by the correct one.
Yours faithfully
Authors